# Seeds of transformative learning and its pedagogical implications on a conference-based university course for environmental and geosciences

Joula Siponen<sup>1,2</sup>, Janne J. Salovaara<sup>1,2</sup>, Karoliina Särkelä<sup>1,3</sup>, Inka Ronkainen<sup>4</sup>, Salla Veijonaho<sup>4</sup>, VeliMatti Vesterinen<sup>4,5</sup>, Isabel C. Barrio<sup>6</sup>, Laura Riuttanen<sup>1</sup>, Katja Anniina Lauri<sup>1</sup>

Box 4, 00014 University of Helsinki, Finland

<sup>&</sup>lt;sup>1</sup>Institute of Atmospheric and Earth system Research (INAR), University of Helsinki, Finland

<sup>&</sup>lt;sup>2</sup>Helsinki Institute of Sustainability Science (HELSUS), University of Helsinki, Finland

<sup>&</sup>lt;sup>3</sup>Water, Energy, Environmental Engineering research unit, Faculty of Technology, University of Oulu, Finland

<sup>&</sup>lt;sup>4</sup>Faculty of Educational Sciences, University of Helsinki, Finland

<sup>10</sup> Department of Chemistry, Faculty of Science, University of Turku, Finland

<sup>&</sup>lt;sup>6</sup>Faculty of Environmental and Forest Sciences, Agricultural University of Iceland, Reykjavík, 112, Iceland Correspondence to: Joula Siponen (joula.siponen@helsinki.fi), Institute for Atmospheric and Earth System Research, P.O.

#### Abstract.









In this study, we explore students' learning experience during a university course in which students studying environmental and geosciences attend the Arctic Circle Assembly conference, introducing them to a wide range of stakeholders and viewpoints from geopolitical to Indigenous perspectives. Using qualitative methods, we studied the students' sense of belonging and transformativeness of the learning process, and how those might influence the development of the students' professional identity. In-situ interviews, written reflections of the students and in-depth interviews post-course reveal elements of the transformative learning process, in which the students' sense of belonging played a role: lack of belonging to the expert community induced dilemmas and belonging to the student group enabled joined reflection. However, some dilemmas do not seem to lead into transformation. Therefore, as pedagogical implications of our findings, we highlight the importance of facilitation of critical reflection and discourse of the learner's values and beliefs. Facilitation should consider students' prior learning and background and include building of trust and belonging in the learning community, enabling the challenging reflections. We suggest that flexible pedagogies and approaches of transformative climate change education have potential to mould students' professional identity and widen their perspectives on what it means to be a responsible scientist or expert in the Arctic context. This requires not only a deep knowledge of the physical processes but also an awareness and understanding of the region's complex socio-economic dynamics.

**Keywords**: transformative learning; sense of belonging; university pedagogy; climate change education; Arctic climate change expertise

#### 1 Introduction

The Arctic climate is warming three times faster than the global average (Zhou et al., 2024), having profound implications for ecosystems, economies, and societies, and requiring immediate action. The region has become an arena for conflicting interests and complicated geopolitical development (Andreeva et al., 2024; Østerud and Hønneland, 2014). Indigenous communities advocate for the protection of their traditional ways of life, cultural heritage, and sustainable use of local resources. Environmental non-governmental organisations call for conservation of vulnerable Arctic nature and biodiversity and for the mitigation of climate change impacts, emphasising the significance of the Arctic to the Earth's climate (Overland et al., 2019). At the same time, nation-states seek to assert sovereignty and secure access to oil, natural gas and rare minerals, as new areas open for resource exploration and transport (Blunden, 2012)—activities that are intrinsically linked to economic systems and extensive exploitation of nature behind climate change. Such contrary motivations are complicated by the current geopolitical tensions, increasing the importance of resolving the issues of Arctic governance, resource exploitation, and responsible use of the environment (Kapsar et al., 2022; Young, 2010). To ensure a sustainable and equitable future for the Arctic, multi-sectoral cooperation is needed. This entails incorporating science, and Indigenous and local knowledge in informing decision-making (Andreeva et al., 2024), which requires transformative changes in institutions, values and cultures (Wheeler et al., 2020). Hence, addressing climate change in the Arctic represents an adaptive challenge rather than purely a technical one, asking for systemic solutions (Gillard et al., 2016), co-learning, as well as willingness to rethink normative assumptions and practices.

Education is a critical tool for both individual and societal transformation, building the necessary capacity to address climate change, and other crises intertwined (Kronlid and Lotz-Sisitka, 2014; Leite, 2024; Mochizuki and Bryan, 2015; Wals and Benavot, 2017). Education can promote climate science literacy (Kubisch et al., 2022), encourage sustainable climate action individually and collectively (Kenis and Mathijs, 2011; Leite, 2024), and strengthen capabilities to innovate, adapt to environmental changes holistically (Krasny and DuBois, 2019), and think autonomously (Mezirow, 1997). To develop such

transformative climate change competence, a transdisciplinary approach that integrates scientific, socio-economic, and contextual dimensions of understanding is needed (Siponen et al., 2024). Being transdisciplinary in climate education is often operationalised in problem- and/or phenomena-based learning, when students are exposed to real-world problem contexts and events.








The role of environmental and geoscience students and experts in this transformative effort is multifaceted. They are involved in bringing complex scientific data as accessible knowledge to wider audiences—bridging the gap between research and public engagement (Illingworth et al., 2018). Through involvement in societal activities, they can help societies adopt systems thinking and make evidence-based decisions (Shrivastava et al., 2023). The experts in Arctic climate change must grasp the interplay between physical processes—such as ice melt and permafrost thaw—and their cascading effects on global climate systems, ecosystems and local livelihoods, as well as the socio-economic challenges, such as the pressures of resource extraction, geopolitical competition, and the rights of Indigenous communities. This interconnectedness of systems, and the need for navigating ethical considerations, requires reflexivity from the practitioners and researchers working in the Arctic. They need to be able to assess their own agency critically, such as by recognising how their work can influence policies, perceptions, and actions within the Arctic. It is evident that geoscientists have a role in informing and engaging in climate action, but competencies supporting this role, such as systems thinking, normative thinking or interpersonal skills, are not necessarily widely incorporated into higher education in the geosciences (Riuttanen et al., 2021). Another critically important, yet overlooked part of climate science education is policy literacy (Selin et al., 2017). Structures, such as the Intergovernmental Panel on Climate Change (IPCC), hold an important position in supporting communication of geosciences to the public and policymaking, and working as a bridge between the science community and stakeholders.

To build the reflexive capacity and transformative competence of future experts and professionals, we need to understand the learning process and what enables transformation better. For this study, we explored students' perceptions of a course called "Arctic Circle", in which they are exposed to and get to engage with real-life (geo)political, environmental, cultural and economic discussions focused on the Arctic. Through interaction with other stakeholders in real-life contexts and the focus on student empowerment, belongingness and active learning, the course represents a unique example of flexible pedagogies in a geoscience context (Matheson and Sutcliffe, 2017; Ryan and Tilbury, 2013). In this learning setting we observe the appearance of two established qualitative theories in educational research: sense of belonging (SoB) (Hagerty et al., 1996) and transformative learning (Mezirow, 2009), aiming at showing their role in the development of university students' Arctic expertise and expert identity. This way we will open a new strand of research and widen our understanding on potentially transformative geoscience learning conditions and environments, breaking barriers of social and natural sciences. Our research questions are:

- 1) What elements of the transformative learning process can be seen in the students' experience, and what aspects of the student were transformed? (RQ1)
- 2) How did the pedagogical approach support the students' expertise development? (RQ2)
- 3) What are the implications to developing education further? (RQ3)

This study conceptualises the transformative learning on our case-course through three recognised stages: 1) realisations and disorienting dilemmas, 2) critical reflection and discourse, and 3) outcomes. With our analysis, our aim was to contribute to the discussion on transformative learning within geo- and environmental sciences, particularly in relation to the use of flexible pedagogical approaches for fostering context-specific climate change expertise.

#### 2 Theoretical background









Transformative learning is described as a process of gaining new insight and models of thinking through facing a disorienting dilemma, reflecting on the new experience and one's relation to it critically, eventually leading into action engagement (Mezirow, 2009). We have chosen to focus on transformative learning, rather than learning in general, specifically to highlight learning outcomes that go beyond informational or instrumental, meaning significant changes in one's ways of being in, relating to and experiencing the world beyond cognitive structures (Hoggan, 2016). As a prerequisite, the learner is expected to have already formed an identity that can then be transformed (Kegan, 2009). These changes can concern a range of aspects, e.g. the learner's worldview, self, epistemology, ontology, behaviour and capacity (Hoggan, 2015). In the context of sustainability (higher) education, we sourced our framing primarily from a review by Rodríguez Aboytes and Barth (2020), who identified a set of possible transformative learning outcomes. These included five themes: 1) increase of new knowledge, understanding of concepts, and practical skills related to sustainability issues, 2) reconstruction of values, norms and perspectives, including increased empathy and sense of interconnectedness, 3) increase of self-awareness, confidence and sense of responsibility that can support agency and willingness to make a change, 4) development of the critical, systems, and complex thinking needed for seeing and approaching the interdisciplinary nature of sustainability issues, and 5) social learning outcomes, such as reinforcement of social relationships between groups and organisations, as well as social action through social mobilisation or activism. These themes represent the need in education for sustainability to engage all aspects of the learner: cognitive/thinking, behavioural/doing and emotional/feeling—in other words, head, hands and heart (Sipos et al., 2008).

To engage the learner holistically and to aim at transformative learning outcomes, the following (simplified) stages were suggested relevant by Rodríguez Aboytes and Barth (2020) according to Mezirow's theory (e.g. Mezirow, 2009): prior learning, disorienting dilemma, critical reflection, discourse, and action. Prior learning refers to the learner's existing frame of reference that is built on previous experiences and learnings, and that selectively shapes the expectations and attitudes that the learner approaches new experiences with and acts (Mezirow, 2009). Disorienting dilemmas appear when new experience doesn't fit within the existing frame of reference of the learner. In this context of a flexible pedagogy course, the dilemmas can be expected to be structured and unintended, meaning that they arise from the nature of the learning activities without the teacher triggering them intentionally (Rodríguez Aboytes and Barth, 2020). To address the dilemmas, critical reflection on the relevant assumptions, perspectives, and information is needed, including one's own critical reflection but also that of others. Communicative discourse has a key role in justifying one's beliefs by seeking consensus with others through understanding their differing points of view and the context of the assumptions (Mezirow, 2009). For the learning to be transformative, the reflection and discourse are expected to lead into the previously mentioned significant changes in the learner's frame of reference, which allow more informed action and critical approaches to new experiences.

As transformative learning requires critical assessment of one's connection to the world and discourse with others in close interaction, we see that the learner's sense of belonging (SoB) can play an important role in the process. SoB is seen as a key personal state resulting from introspective realisation of feelings of being an (integral) part of the surrounding system or environment (Hagerty et al., 1996). In our case, the system and the environment include 1) a course, called "Arctic Circle", which utilises a conference as a learning activity, and 2) the related groups of people. SoB connects to a sense of purpose and meaning, and in a learning context it can manifest in students feeling accepted, respected, included, or supported by others in the social environment (Allen et al., 2021; Goodenow and Grady, 1993). SoB shapes students' holistic experience and has been shown to influence retention and overall success in studies (Thomas, 2012). Students who feel belonging are more active

learners, and experience learning as being easier and more enjoyable (Kahu et al., 2022). In addition, SoB in science, technology, engineering and mathematics (STEM) studies has been shown to support career interests (Xu and Lastrapes, 2022).




To study and promote students' SoB, several frameworks aim to distinguish between aspects of the learning context that influence a student's belonging. Various domains have been suggested as relevant for SoB, including academic sphere, social environment, social participation and physical environment (Cohen and Viola, 2022), or academic and social engagement, surroundings and personal space (Ahn and Davis, 2020). Belonging in an academic setting can be influenced by participation in the teaching situations, self-efficacy and interest in the study subject and feeling that it is relevant to one's own career path (Kahu et al., 2022). Familiarity with the subjects studied, or the people around is a precursor for belonging (Kahu et al., 2022). Naylor (2017) also showed that a chance to personalise one's studying to be meaningful to oneself can create positive academic belonging. The physical environment or surroundings as living space or geographical and cultural location can be an important aspect of belonging through attachment and emotional connection to the place.

To increase individual and collective SoB, Allen et al. (2021) suggested that four elements should be considered: motivation, competencies, perceptions and opportunities for belonging. Lacking skills, motivation and/or chances to belong, can create feelings of alienation, powerlessness and meaninglessness that then hinder learning, in addition to having negative health effects. Plenty can be done along these lines in terms of course and curriculum design to support the students' SoB in when studying. As the students' expertise develops, and they start to build their professional identity inside or outside academia, the targets of SoB transform as well, from being a study and learning community (Myyry et al., 2024) to becoming a community of practice (Jackson, 2016; Wegner, 1998). Attending an event like the Arctic Circle Assembly can be an important experience for exposing students to this transition and starting to build their identity as an expert.

Identity is exactly what is transformed in transformative learning, according to Illeris (2014). Identity is tied to one's perception of the groups they belong to, hence also to their SoB. Gjøtterud and Krogh (2017, 9) stated that "transformative learning originates from intersubjectivity and humans' bodily relations to the world". In this intersection of social relations and their reflection seems to lie a set of keys to transformative learning experiences—from the disorienting dilemmas that may rise as one reflects their belonging to a context, to the pedagogical spaces that develop and nurture belonging, trust (Burke et al., 2016; Matheson and Sutcliffe, 2017) and positive social identity in the learning community (Myyry et al., 2024), to transforming the very identity.

In this flexible pedagogy study context, attending the Arctic Circle Assembly as a learning activity may shake the learners' identity and SoB. In the event, students of geo- and environmental sciences are actors in close and active interaction with the socio-cultural context in the Arctic. In this process, they develop their Arctic expertise by being exposed to an expert community and a context outside their own academic field of study. We have recognised that the literature on interdisciplinary courses utilising expert events as well as studies on the interconnectedness of SoB and transformative learning is limited.

Therefore, we want to widen the discussion by exploring the students' learning experience in this specific higher education setting.

#### 3 Methodology







#### 3.1 "Arctic Circle" course

The "Arctic Circle" course lends its name from an annual event to which the students participated in the course. Annually, the Arctic Circle Assembly welcomes a multisectoral audience of approx. 2,000 participants, as stakeholders of Arctic issues, to discuss: geopolitics and security, climate change, Indigenous perspectives and rights, economic and sustainable development, as well as scientific research and innovation. The conference is held in Reykjavík, Iceland, comprising a full three-day programme of plenary and topic-specific sessions to receptions, video screenings and round table discussions: "Arctic Circle is the largest network of international dialogue and cooperation on the future of the Arctic and our Planet. It is an open democratic platform with participation from governments, organisations, corporations, universities, think tanks, environmental associations, Indigenous communities, concerned citizens, and others" (The Arctic Circle Foundation). Historically, the Arctic Circle network was built on political grounds in 2019 and led by the former president of Iceland Ólafur Ragnar Grímsson.

The "Arctic Circle" course is a master's level course given by the Agricultural University of Iceland. The course was originally designed specifically to be part of a Nordic Masters' Programme in Environmental Change in Higher Latitudes (EnCHiL), during which students study geosciences, environmental sciences, and social sciences. The EnCHiL programme does not specifically address themes related to politics and business that are visible in the Arctic Circle Assembly. Therefore, this course was developed to expose students to a side of Arctic sciences that they might not yet be familiar with, and to reflect on their own role and identity.

The learning outcomes, respective learning activities and assessment tasks of the course are shown in Table 1 and the course timeline in Figure 1. One in-person session was held to prepare the students for attending the assembly. One of the teachers of the course has previously been involved in organising the conference, and thus was able to provide an empirical perspective of the event and offer some guidance on navigating the programme to find suitable sessions. Before the conference, each student presented their participation plans in an online session, allowing their peers to be motivated by the topics and activities chosen by fellow students. After the conference, the students were given 1-2 weeks to reflect on the experience before a joint discussion session was held. The course concluded with the students submitting a report on one topic presented at the assembly and a reflection on their overall experience of the event. (Assignment instructions are included in the Appendix A). The teacher facilitated the learning sessions and interacted with the students actively during the sessions and the event. The course group was small: 13 students in total, with many students being familiar with the teacher from previous courses. During the conference, the students were accommodated in the same location for two nights, provided by the university.

The pedagogical approaches of this course can be seen to reflect transformative climate change education, as Leite (2024) framed it, even though transformative learning was not an explicit goal of the course design, and rarely is. The aspects of Leite's framework that align with the course include:

- transdisciplinary and real-world connections: students interact with real-world political, social and economic contexts in the event through the course report and networking exercises
- cultural responsiveness: students are exposed to and engage with a range of cultures and knowledge systems,
   including Indigenous perspectives
- collaboration: the course includes active discourse between students and teachers prior, during, and after the event
- phenomena- and place-based: in their assignment, students work on a theme of their choice related to the Arctic, considering various perspectives, and searching information independently.

Table 1 "Arctic Circle" course structure: intended learning outcomes, learning and teaching activities as well as assessment tasks

| Intended learning outcomes |                                                                                                                                                                                                                                                                                                            | Learning/teaching activities |                                                                                                                                                                                                 | Assessment tasks |                                                                                                                                                                   |
|----------------------------|------------------------------------------------------------------------------------------------------------------------------------------------------------------------------------------------------------------------------------------------------------------------------------------------------------|------------------------------|-------------------------------------------------------------------------------------------------------------------------------------------------------------------------------------------------|------------------|-------------------------------------------------------------------------------------------------------------------------------------------------------------------|
| 2)                         | Gain knowledge through various academic fields that contribute to Arctic studies and increase the depth and understanding in their field of specialisation  Have knowledge and understanding of the role and interaction between stakeholders, including local and national governments,                   | 2)                           | An in-person introduction session before the Assembly (2 teachers present)  An online session before the Assembly for students to present their plans (1 teacher present) + introduction to the | 2)               | The course report (70% of the grade)— an academic essay on a topic of their choice A networking exercise (20%) in                                                 |
| 3)                         | private businesses, non-governmental organisations and the public  Understand the relevance of economic and social issues in the Arctic  Be able to evaluate the suitability of the different methods of analysis and complex scientific issues for research and decision-making related to Arctic studies | 3)<br>4)<br>5)               | research  Active participation at the Assembly  In-person discussion session after the Assembly  Writing a reflection on the Assembly and own experience & a report on one session topic        | 3)               | which they must<br>interact with at least<br>two other<br>participants at the<br>conference<br>A reflection exercise<br>(10% not graded, but<br>just "pass/fail") |

Figure 1 Course timeline with data collection

# 3.2 Materials and sample



To gain insight on the learning experiences of the students, we employed exploratory qualitative methodology, focusing on the individual students' reflections. We collected data in the following ways (see Figure 1): semi-structured interviews in English during the event (N = 9 out of 13 students on the course); written reflections (N = 9); and hour-long semi-structured interviews post-course (N = 4), aiming for depth rather than width in the material. The study was introduced to the students in the online session before the event and their consent to participate was confirmed. In the mandatory course assignment, the students were instructed to choose what they reflected on from the following prompts: what they learned; what they enjoyed; whether they experienced interaction as being challenging, and whether they gained a broader understanding on Arctic issues (see Assignments in Appendix A). Most students reflected on all of them, only two leaving out reflection on challenges in

interaction. The course teacher collected the written reflections, when they were submitted, and shared them with the research team.

Interviews at the event focused on the students' SoB in the event and on the course, and how they felt about their own expertise in the context of the event. The *in-situ* interviews were conducted by three of the co-authors, and they were recorded and transcribed. The *post-fact* in-depth interviews conducted by the first author focused on the students' learnings and possible transformations. Therefore, they were done one to two months after the final deadlines on the course to allow the students' experience to simmer and possible transformations to start finding their shape. While the initial interviews were conducted on-site at the event, the post-course interviews were conducted via Zoom as the researchers and students were in different geographical locations. While this could be seen to limit the richness of the non-verbal communication, the students have had online education in their programmes as well as during the COVID-19 pandemic and therefore are used to communicating via Zoom. Also, the post-course interviews were recorded and transcribed, and extensive interview notes (Bryman, 2016) were taken by the interviewer. The guiding questions of both sets of semi-structured interviews (King and Horrocks, 2010) are included in Appendix B.

The participants in the study were a mixed group of students. Five bachelor's students and one master's degree student were studying in Iceland as exchange students. Two students were taking part in EnCHiL, the Nordic Masters' programme. One student lived in Iceland and was studying in a master's level programme there. According to their own record, the students represent the following fields of study: forest and nature conservation, forestry and landscape architecture, ecological sciences, governmental systems, natural/environmental sciences, landscape construction and environmental science and law, as well as environmental and natural resource economics. From the post-interviewed students, two were bachelor's degree students and two were in master's degrees.

## 3.3 Analysis process









The entirety of the collected material, including transcripts of interview recordings and written reflections of the students as documents, was subjected to qualitative content analysis (Bryman, 2016), utilising Atlas.ti software. The material was initially coded by the first author, who is an outsider to the course and was not present during the event. Initial coding was done inductively, aiming for new insights in the context to emerge (Elo and Kyngäs, 2008)—which, importantly, led to the realised connection between the transformative learning process and SoB. The coding was then further matured (i.e., further organised by reappearances (Krippendorff, 2019), deductively based on the existing theory of transformative learning. Its stages were specifically recognised: disorienting dilemmas or realisations, critical reflection and discourse, and the outcomes as a transformed frame of reference (Rodríguez Aboytes and Barth, 2020; Mezirow, 2009). In addition, its connection to the students' SoB and where it interacted with the learning experience were examined. Some of the initial inductive codes remained under these newly introduced, higher level categories and some were combined, renamed, or deleted according to their relevance. The coding results were discussed and reflected on within the co-authoring group at multiple iterations along the analysis process, and the results found their final form in the writing process. The analysis is influenced by the existing experiences and knowledge of us authors, consisting of backgrounds in education, as well as environmental and atmospheric sciences. The authors have recently studied SoB in the context of environmental and atmospheric sciences education, to which this study acts as a continuum. In addition, as one can argue that the transformative learning process is only relevant as a whole and as a dynamic, ever-changing and continuous process (e.g. Mezirow, 2009) and therefore nearly impossible to unambiguously analyse from the material at hand, we have focused on pointing out elements of it and their interactions with the students' descriptions of their SoB. The direct quotations of the students are used to support the transparency, rigour and credibility of our results (Lincoln and Guba, 1985; Elo and Kyngäs, 2008). In the Results, the students are referred to by IDs AC1 to AC9, and the materials as 'pre', 'refl' and 'post' referring to the interviews in the event, written reflections and post-course interviews, respectively.

#### 255 4 Results





Students described several realisations leading to disorienting dilemmas, reflections based on those realisations, as well as resulting transformations in their frame of reference. The presentation of results in the following subsections is based on these three steps of transformative learning.

#### 4.1 *In-situ* realisations to disorienting dilemmas

The students' realisations emerged from how they related to the content of the event, or to other participants, or to the political or cultural impression. The realisations reflected contradictions that the students experienced regarding ideological standpoints, intentions, ways of communicating, attitudes, and perspectives. The "real world" portrayed during the Arctic Circle Assembly, differed from the students' expectations and dispositions, that is, their existing frame of reference.

An important source of the dilemmas was the lack of SoB experienced, especially to the expert community. The students described their SoB in general as being influenced by the lack of familiarity to the people, topics and the cultural context of the event—how they perceived themselves fitting or adapting to the learning context. Most students felt they were primary members of the student group, and only a few of them felt they belonged also to the expert community. For example, students felt *intimidated* by the professionals who seemed to have strong networks at the event. AC2 described their feelings as follows:

"I did not really feel like I belonged there, because it was just a bunch of businessmen in suits walking around and meeting their colleagues, [...] or science people who had a lot of knowledge, and I was just feeling like a little student being there, not knowing a lot of stuff and just curious to learn new things" (AC2 post)

However, some students felt more belonging to the event. Some thought themselves to be part of the researchers' and scientists' community because of their assumed shared university background, while AC7, for example, felt that they belonged to the event because of their shared interests:

"I just feel like it's a lot of individuals who can do many things together, and therefore I feel like I belong here [...] I think [I belong to] young environmentalists that need actions because of their future. [...] I feel like I belong in assemblies related to the climate crisis and environmental matters, because I have a bachelor's degree in that matter and most of my time goes to something connected with that" (AC7 pre)

They even thought they could have hosted a session at the event, suggesting that they saw themselves as being equal to the other participants. In general, the students' perception of their expertise and relation to the expert community relied on the breadth and width of their knowledge, as expressed in statements such as: "I feel like I don't have the knowledge that the others have" (AC2 pre) or "I'm interested in the stuff but I do not know a lot about it yet" (AC3 pre), or as one student described: "Engaging with other participants proved to be challenging for me, as I felt like more of a listener and observer throughout the conference—rather than a 'true' participant who could contribute" (AC6 refl).

This perception caused students to limit their engagement with others than their peers: "Many of us struggled to interact with other participants because we simply didn't really know how to and felt that we 'didn't have anything to offer' for the participants we were interested in" (AC9 refl). They also stated that they have the right knowledge, but not the *expertise* yet, seeing themselves as experts in the future. One saw the event as an important learning experience towards expertise:








"I don't feel like an expert because [...] I'm still student, but I'm sure that attending to the sessions and listen to people at speaking to people I learned a lot more, so I'm not an expert but I'm sure my knowledge is bigger, [...] way more than before" (AC5 post)

How specifically the students saw themselves as belonging to the event was evidently influenced by their SoB to the other participants and the contents. The students stated that they had expected more scientific content, discussions on nature, strategies to tackle climate change, and opportunities to gain new knowledge. Instead, they perceived the conference as mainly focusing on business and politics, and somewhat disregarding the attention to specific measures of solving Arctic challenges. This perception was shared, in one way or another, by every student. For example:

"We were expecting something [...] something more on the scientific environmental side, and we didn't expect so much economy, so much geopolitics [...] even though it's important. And we didn't expect so much greenwashing. I mean it was really visible, nobody cared, and we were like 'ah, okay that's the thing'. That surprised us" (AC5 post)

Others described similarly: "I had hoped that I would learn a lot about new scientific results [...] however, the sessions mostly talked about all the problems we already know [...] or new business ideas" (AC3 refl), or: "[...] after all those sessions I was still realising that there's something going really wrong in the behaviour of people in general with all that wealth and the economics actions" (AC8 refl).

The realisation of conflicts in values, regarding themes such as vague political talk, false advertising and visible greenwashing, left the students feeling disappointed or even sad. At the same time, they gained new insights, as stated by AC7:

"The Assembly did not provide me with a broader understanding of current environmental issues in the Arctic, but it did open my eyes towards the political challenges of the Arctic: for example, how sailing will become different with no ice in the Arctic. And I found it weird that sometimes they spoke about the opportunities of climate change" (AC7 refl)

Another often-mentioned topic that evoked feelings and realisations among the students was the issue of representation of different groups at the event, such as Indigenous peoples or youth—as the future stakeholders, actors and decision makers. To most students, issues related to Indigenous perspectives were new. They felt that the representatives of Indigenous peoples in the Arctic and young people, while invited to the event seemingly equally, didn't get their voice heard as much as those talking about economy and (geo)politics. At the same time, the students found the presentations and talks from the Indigenous viewpoints to be the most meaningful and thought-provoking: "the Arctic Circle Assembly was not really doing what they advocate and advertise for, it was just kind of: 'yeah we're doing these talks, and we're giving people voice' but then actually they kind of didn't" (AC2 post). Again, this felt like a disappointment and a contradiction of values:

"I do think it is important to acknowledge that the event is still mainly about money and power. There are constant expressions that radical change is needed, but no one is really performing this change. There is recognition of





indigenous voices to be heard, but indigenous voices were still in the background of the Assembly. There is mention of youth in some sessions, but the young are still not being listened to" (AC1 refl)

Here too, while these contradictions felt somewhat personal and, in a manner, unsolvable, many of the students mentioned the importance of being present at the event. They were able to learn through personal stories and viewpoints empirically, whether through viewing a documentary movie, hearing a presentation at the event, or simply experiencing the contradictions and challenges other people seemed to face in such a global arena:

"[...] I already knew before that there are some issues related to the indigenous communities that they don't really have a voice and that they are being suppressed and these topics were brought up in the Arctic Circle again, so for the first time I could see with my own eyes how it is and not just on Instagram or on social media and actually hearing them talking by themselves about this stuff was different and then also to see that the issues are real, and it's even in the Arctic Circle Assembly [...]" (AC2 post)

Altogether, experiencing these topics that many of the students were familiar with, but in a new real-world context, was often thought of as very impactful: "[...] the Assembly allowed me to understand the Arctic better and the many different perspectives helped a great deal. [personal stories] enriched my understanding of what problems people are facing in the Arctic" (AC1 refl), and: "[...] a presentation that left me with a lasting impression and kept me thinking was a screening of a film followed by a discussion. [...] I wasn't aware of the current situation of the Inuit and how they continue to suffer and be oppressed [...]" (AC6 refl). The same sentiment was felt throughout different represented groups: "I was really amazed because it was talking about the situation of Indigenous people in Greenland [...]" and "In one session there was also a kind of exchange of comments between a Maori, explaining the ecosystem services with the sea in the Pacific Ocean, and a whaler from Iceland" (AC5 post).

# 4.2 Reflecting to settle the dilemmas

Various and numerous mentions of self-reflection, of differing degrees, were present in the students' descriptions of the event and the course. For example, AC1 reflected on their own motivation, agency and future responsibility as a researcher, especially considering the people and contradictions in the Arctic; while AC2 reflected on the role of young people in the future and how their engagements had developed their thinking to be more critical; and AC5 reflected on their broadened awareness from exposure to Indigenous and other cultural representations. AC8 reflected these in relation to their belongingness, saying:

"I think it was not a bad thing that I didn't have the feeling of belonging there but [instead] it was a good thing to get out of [my] box and that helps me a lot, it was a memorable experience" (AC8 post)

The students recognised the importance of their peer student group for their learning and reflection. SoB to the group played a major part in discourse and critical reflection. The students reflected on their SoB stemming from the various elements that they shared with the students, and to an extent, with the other participants of the event. These elements included aims, interests, values and frustrations, but also the physical space, the experience and joint reflections. AC1 stated that they were glad to have shared the experience of reflecting on their own agency, responsibility and, also, the right to be part of an event in the Arctic with the other students, and elaborated on how it brings hope:








"It's very nice to know you're in the same boat, you know? You're just students and you are there because the opportunity was presented to you and you said yes [...] you realise that things aren't going the way they maybe should, but then it's really nice to realise that these people of your generation are thinking the same way" (AC1 post)

Also, personality traits and interpersonal skills of the students are a factor. For example, while having challenges in interacting with other participants, AC2 was still happy about the opportunity to attend the event with people that they already knew and felt that interacting with the other participants was easier together with the other students, whereas AC5 didn't feel a strong connection to other students. AC6 and AC8 found peer interaction challenging, but still felt belonging to their group, and both were okay to attend more alone, and AC4 wanted to focus more on the content anyways. Altogether, a lack of SoB seemed to negatively influence engaging with other participants.

For AC2, the joint discussion and reflection with other students had a major influence on how they related to the event: "the more I talked with the other students about their experiences the more I kind of understood what is going wrong here" or to AC5: "we had kind of the same ideas, [...] someone more extreme, someone more in the middle way but the idea is, we were kind of in this same area". Thus, realising that everyone is on the same page can be an influential experience. This highlights how the reflective learning experience depends on the students' prior existing frame of reference and on SoB the student has to their group, through shared understanding and processing of the experience.

The students described gaining important new insight concerning how money and power act as the main forces of action, and that there is a lack of actionable plan, but also how some are actually working for change, as for example AC5 stated: "I acquired insight into the challenges in making decisions to safeguard the environment, the political landscape, and the socioeconomic state of this crucial region of the world. It necessitates collaboration with a multitude of people from across the globe" (AC5 refl). While plenty of the reflected learning felt important and meaningful, they were harder to link to critical reflection and significant changes in the students' frame of reference. Some of the realisations and conflicts of values that the students experienced seemed to be disorienting to the students and as such had an impact on their learning outcomes, appearing as transformative learning.

## 4.3 Learning from the disorienting dilemmas

In our search for transformative learning, we aimed to explore the course outcomes to the *aspects of the learner that change* (Hoggan 2016), e.g., the student's epistemology, worldview, expert identity, and themselves in general. Some also recognised changes in their behaviour. While many of the learning outcomes, described below, are connected with the dilemmas presented earlier, it is indeed tricky to distinguish whether learning happens at the initial realisation leading to reflection, or is a result of the reflection—acknowledging that in cyclical and dynamic learning processes, both are encompassed.

Students recognised gaining a broader perspective to the Arctic's socio-economic situations and challenges, and matters related to Indigenous peoples and perspectives—that seems to have led to changes in the students' worldviews. For instance, insight into the Indigenous communities, their way of living and thinking, were new to many students, who later recognised that these were perhaps the most important learning outcomes of the experience:

"I knew about Indigenous knowledge [...] but I attended a couple of sessions about [it] and that opened my mind about this topic even more, the importance of it. It expanded my knowledge about that and my conscience" (AC5 pre)

AC1 reflected: "there was so much inequality and still so much governance and policies that just contradict [the Indigenous] way of life, and being able to realise that was very important" (AC1 post).







In connection to feeling things represented unequally at the event, another realisation that shifted the students' thinking was the observed lack of action or viable action plans for tackling the commonly recognised challenges. AC2 concluded that in their view, the whole point of the event was "to make business with each other but not really actually changing something" (AC2 post). This made AC7 feel disappointed and quite hopeless: "I would like to improve what way we speak about actions and what point we want to reach, not just where we are now [...] I don't know how much we're gonna get from [such] assemblies" (AC7 pre). Students found their role (and agency) quite difficult against the large institutions most vocal and visible at the assembly:

"I'm getting used to the way that those things go [...] it's still some institution that's kind of relying on oppression and exploitation and then it's just kind of the way things go now. And you hope it's going to change in the future, and you really hope that your generation is going to do differently but there's so much uncertainty in that as well so yeah, there's a sense of hopelessness and a sense that I am not able to change anything" (AC1 post)

Many students thought that even if you would not be able to change things, it is important to fully acknowledge the situation. The shared frustration was something that brought the students together and heightened their SoB and opportunities for reflection. AC2 described how their critical thinking was developed during the course: "I think I would look more critically at what they're actually doing [in an event like this]" (AC2 post) and AC7 reflected on the importance of critical thinking: "these assemblies should be used in a specific way for a more positive and meaningful outcome [...] I learned how important it would be to have a critical eye when attending these assemblies" (AC7 refl).

Regardless of their critical views, many students acknowledged the importance of understanding familiar issues such as climate change and its global scale in this real-world contextual situation (e.g., the Arctic geopolitical landscape):

"that was really interesting to have a little [sneak] peek to the rest of the world [outside] the research world [...] to learn more [about geopolitics] [...] if you want to study something like what I'm studying right now, I think it's really important to know every aspect of the issue of climate change in the Arctic" (AC5 post)

AC8 said: "it was really nice to get a feeling about the challenges and problems the nations in the Arctic circle faced in the past, present and future" (AC8 refl).

Some also thought the event to be an "eye-opener towards the important topic of youth engagement" (AC7 refl). Students underlined how they felt about the importance of listening to diverse (e.g. Indigenous and young) voices in decision-making and in science and in general the necessity of collaboration for finding solutions. Students connected this new awareness of diverse perspectives to their disciplines and future work as Arctic experts—hence, to themselves:

"[...] if I were ever to work on any conservation, at least I am able to recognise that my way of thinking is not going to be the right way—that I really need to listen to people and that is regardless of where you are globally that you really need to listen to what people's needs are" and: "you forget that there are still people that live there and have their entire lives there and that they will always be there [...] as a scientist, being able to give back to communities, you know, to share your results and to help their way of life and their livelihood." (AC1 post)

In a similar vein, students related to the personal stories and recognised the importance of plural perspective in solving these issues, suggesting a more open and reflexive epistemology:

"you're always looking for the best results and numbers and stuff, but then we fail to acknowledge how important it is that we're still just telling our story, and science is just a tool in the greater kind of an idea" (AC1 post)

Awareness of different ways of knowing and the oppression experienced by Arctic people seemed to lead into a new sense of humility and agency in how the students would approach doing science in the future: "we are in the Arctic, but we are raising [collaboration] to a global level, you know? [...] the whole time I was there, was like: who am I to do anything here? But I do realise that I am someone, and I do, yeah, I do matter in a sense" (AC1 post).

Some gained clarity in their career goals by finding new research needs as described by AC5, or becoming more familiar with the existing research community that the students aspired to become a part of, as AC2 reflected on: "It was looking up to them. Like okay, they really know their stuff and at some point in my life I will hopefully also know that much." (AC2 post)

Lastly, the students' newly gained insight and skills were taken as incentives to act and to make changes in their behaviour. As one student stated: "I acquired knowledge in how to communicate effectively with people from diverse professions and realities, along with those who share my field of study" (AC5 refl), and in relation to their studies another described how: "The Assembly motivated me to work hard to someday be part of it and be able to see actual changes" (AC9 refl). The students also saw their attained knowledge and experience as being something that was worth sharing with other students, not on the course:

"I can spread my knowledge to my friends or if there's a discussion, I can bring up my experiences from the Arctic Circle" (AC2 post). However, not all felt the experience to be worth the contradictions, as one concluded: "I will not recommend this to anyone since this is nothing more than people trying to sell their greenwashing ideas" (AC4 refl).

#### 5 Discussion and conclusions






Sourced from reflective interviews, our research design of qualitative exploration into students' learning experience on a real-world contextualised course led us to several novel findings. The "Arctic Circle" course appears as an example of a transdisciplinary and possibly transformative higher education climate change learning experience for students of geo- and environmental sciences (Leite, 2024). Evidence of the transformative learning process was found through the analysis of the interviews and the written reflections of the students (RQ1). The lack of belonging and contradictions in values and representations of the Arctic pluralities were sources of dilemmas that were then reflected on amongst the student group. The sense of belonging and personality traits played a role in the discourse, which then led to widening the perspectives and increasing the awareness of the students on Arctic issues, as well as them becoming responsible (future) Arctic experts. The students' learning experiences have pedagogical implications with which transformative learning in higher education could be supported, as well as rhetorical implications regarding the role of experts acting for climate change (RQ2-3).

By facing the plural socio-economic realities of the Arctic, evident in the Arctic Circle Assembly, the students had impactful realisations regarding the content of the event and their own role and place there. These realisations can be seen as disorienting dilemmas: as experiences conflicting with the students' existing frame of reference (Mezirow, 2009). Students described their expectations not being met through emotions such as disappointment, hopelessness and sadness, indicating, for example, experienced conflicts of values and approaches to issues important to them. Critical reflection and management of the emerging dilemmas, alone and together with the learning community, is a key for their frame of reference being potentially transformed,

and for their individual identity to develop (Matheson and Sutcliffe, 2017). The students' descriptions did not always show direct causality from dilemmas to transformative outcomes, but rather to attaching new meanings or perspectives to their existing understandings (Leite, 2024). It has been said that transformative learning has emerged from the interaction between the learner's state of readiness, and the quality of the learning environment (Sterling, 2011). Therefore, as a reaction to flexible pedagogy, teacher facilitation could be strengthened to encourage the students to face the emerging dilemmas and to reflect critically on their own values, beliefs and feelings related to them. This facilitation should be also based on the learners: such as their background education and attitude about learning. In addition, personal capabilities and characteristics have been shown to influence the (transformative) learning experience, through varying capabilities for interaction, or skills for reflection. Facilitating self-reflection and discussions with peers on students' existing knowledge, perspectives and expectations prior to the conference, and learning experiences alike, could help students to self-regulate their learning experience better. Here, utilising the diversity of the student group in discussions to pluralise their perspectives, has been shown beneficial (Matheson and Sutcliffe, 2017).

Based on the materials, SoB—or the lack of it—appears to have an important role in inducing dilemmas that can disorient. SoB was also an important supporting element in the students' shared discussions on their own values and expertise development. However, it is useful to recognise that in some cases, seeking belonging might lead to avoiding conflict or disagreement. This could be seen as a risk to further polarisation (*us* vs. *them*) between the students and parts of the assembly. Again, the facilitation of critical reflection on students' own values and beliefs is crucial, and requires building a learning environment of trust, where disagreements and differing perspectives can be voiced and discussed safely (Matheson and Sutcliffe, 2017). In an ideal situation, critical reflection and constructive discourse would lead to an increased sense of agency, opposed to hopelessness, both visible in the students' contemplations on their learning in this setting. We see here a potential implication to geoscience education: on how to integrate transformative learning, transdisciplinary and flexible pedagogies in their teaching and higher education curricula. Leite (2024) proposes action-oriented and community-based approaches as a medicine to hopelessness at the face of the "unsolvable" climate change. In this learning setting, engagement in action with one's own hands could mean that participation in the event with their own session, as one student suggested (Mezirow, 2009; Sipos et al., 2008). In addition, the strong feelings that emerge when facing contradictions—the emotional aspects—could be approached through artistic or other creative methods (Leite, 2024), perhaps connecting to the formats present in the Arctic Circle Assembly, such as film.

The exposure to the plurality of Arctic issues and perspectives seems to have resulted in widened views and an increased understanding of the setting that many of the students see themselves working in as they transition towards being Arctic experts. At the conference, the students were able to associate with a community of practice (Wegner, 1998), and some felt that they had agency in this context, while also their interpersonal and communication skills improved. They saw their future role as Arctic researchers with responsibility over how science should be done: taking people into account by listening to differing voices; being open to various ways of knowing; and by critically assessing their own and surrounding normativities. Thus, this experience as a flexible pedagogy course can also support geo- and environmental science students' career planning, and the development of their multidisciplinary professional identity and transferable work-life competencies. From the perspective of geoscience education and professionalism, building a bridge between multisectoral arenas—such as the Arctic Circle Assembly—has clear benefits for both the communication and impact of geosciences in the Arctic (Illingworth et al., 2018). Better awareness of not only the environment, but also the people who live in it, the economic ties that complicate it, or the geopolitical tensions that complexify it, are vital viewpoints of the Arctic context, when the actor's aim is to contribute to solving real-world challenges. For now, the real-world context has perhaps felt detached from the students' experiences,

and as this study shows, exposure outside the academic ideals proved fruitful for the students' professional identity development through transformative learning.

We acknowledge that this study focuses on a single course in a programme with a relatively small cohort of students, which limits the generalisability of our findings in more universal educational contexts. However, our methodology and case selections were made with to the aim of collecting and reporting the various divergent experiences of the students, rather than quantifying or ranking the most frequent or impactful experiences. Additionally, the students' prior knowledge was not measured pre-course, making it more difficult to assess isolated changes or transformations in other means but student reflections and theoretical hypotheses (Hoggan, 2016). Related to this, as the course was conducted over an intensive short period, potential long-term impacts of the learning experience remain a question as is typical in such research. Future research should further investigate transformative learning in flexible pedagogies of transdisciplinary settings, specifically focusing on their prior frame of reference, to provide more understanding of the interplay between SoB and transformative education.

In conclusion, we recognise the transformative potential of the "Arctic Circle" course as a flexible pedagogy learning setting, in which the students are exposed to the plurality of Arctic issues and perspectives, fostering a deeper understanding as well as a sense of responsibility as the Arctic scientists of the future. The students' reflections revealed a disconnect with the values and representations in the expert community presented by the Arctic Circle Assembly. Perhaps, the goal of higher education, including in the geosciences, should not merely be to mould the students into the existing community of practice, but also to empower them to assess and reshape the structures around them critically, building communities that reflect their own values and support their agency as contextual climate change experts. Supporting the development of such competence, the benefits of transformative, transdisciplinary, and flexible pedagogies put the focus on facilitating critical reflection and on building a learning community of trust and belongingness.

#### 520 Author contribution statement





SJ, SK, SJJ, LKA, RI, VVM, and VS participated in the conceptualisation of the study and methodology. BIC taught the course and provided information on the course structure and pedagogies. SK, RI and VS conducted interviews at the event. SJ, SK and SJJ planned the post-course interviews with comments from other co-authors. SJ conducted post-interviews and the initial content analysis. Second round of analysis was done with comments from SJJ, VVM, LKA and RI. SJ prepared the original manuscript draft with contributions from SJJ, BIC, LKA and RL. All co-authors reviewed, edited or commented on the final manuscript. LKA acquired funding for the ABS We Belong project and RL for the ClimComp project.

## **Conflicting interests**

The authors declare that they have no conflict of interest.

## **Ethical statement**

This study was performed under ethical guidelines from the University of Helsinki. However, explicit review by the ethical research board for this study was not mandatory.

#### Acknowledgements



The authors wish to thank the students of the course who gave their valuable time and shared their learning experience for the purpose of research. This study would have not been possible without them! We want to thank the Agricultural University of Iceland for organising the course, and Arctic Circle for the assembly that has proved a fruitful learning setting. We also want to acknowledge the editor and the editorial office of the journal as well as reviewers for their time and effort to improve this manuscript. Finally, thank you to Ian Dobson from the Language services of the University of Helsinki for the language revision and important corrections.

ChatGPT has been utilised when summarising themes of the conference programme, searching for relevant literature, and for text editing on a sentence level. Avidnote was used to search for relevant literature.

#### Financial support

This study was based on work funded by the Nordplus Higher Education framework of the Nordic Council of Ministers, under grant: NPHE-2023/10209 (Network: "ABS - Atmosphere-Biosphere Studies", Project: "We belong – contributing to a sustainable development of international education"). In addition, it was supported by the Research Council of Finland grant: 340791 and 340794 ("Learning of the competencies of effective climate change mitigation and adaptation in the education system (ClimComp)").

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

#### **Appendices**

#### Appendix A





# 655 Course description (given to the students)

The Arctic is experiencing accelerated changes. The economic and sociopolitical relevance of the Arctic is expected to become more important soon as climate change makes natural resources and transport routes more accessible. While these changes will create economic opportunities, they also pose threats to fragile ecosystems and societies. The extent and the thickness of sea ice in the Arctic have decreased dramatically in the last decades, especially during the summer months. Sea ice melting facilitates natural resource exploration in the high north, which is estimated to host 13% of the world's undiscovered oil and 30% of undiscovered natural gas reserves. Moreover, the retreating and thinning of the ice opens new trade routes.

In this course students will participate in an international meeting, the Arctic Circle Assembly (http://www.arcticcircle.org/), that takes place annually in October in Harpa, Reykjavik. During the meeting students are required to attend at least three sessions on topics of their choice and will write a report after the meeting. In addition, students will attend two classes, one shortly before and one after the assembly.

**Description of assessment:** Evaluation will be based on written reports (three assignments) in which the students summarise and discuss one of the topics presented at one of the three sessions attended, reflect on their participation in the assembly and develop their networking skills. Students must send in their reports no later than four weeks after the Arctic Circle Assembly.

#### **Assignments**

#### Reflection on Arctic Circle (10% of final grade)

You will have to provide a short reflection on your participation in the Arctic Circle Assembly 2023. Please comment on one or several of the following questions:

- what have you learned by attending the Arctic Circle Assembly?
- had you attended this type of events before?
- what did you like best/worst of the event?
- did you find it challenging to interact with other assembly participants?
- did the assembly provide you a broader understanding of current issues in the Arctic?

# Networking exercise (20% of final grade)

Conferences are important, not only because of the transfer of knowledge, but also because of networking opportunities. One of your assignments during the Arctic Circle Assembly 2023 will be to mingle during the coffee breaks and to meet some new people. For this assignment, you will have to talk to at least two people (including if possible one speaker at

the conference). You will have to introduce yourself, and ask them the following questions (or something along these lines):

- 1) What is their name and where do they come from?
- 2) Did they give an oral presentation at the conference?
- 3) Is this their first time attending the Arctic Circle Assembly?
- 4) How does their work or daily life relate to the Arctic?

#### Report of session (70% of final grade)






During the Arctic Circle Assembly you will have to attend three sessions (including one plenary session). You will have to choose one of the sessions as the topic to write a short report. You can choose if you want to focus your report on the specific topic discussed during the session, or some issue related to the session's topic that was brought up during the discussions in the session.

The report should be 1,500-2,000 words long (without the list of references). You will have to use relevant references where needed to support your statements. Please, include all references cited in the text in a list of references at the end of the document (this list of references does not count towards the word limit); if you have any questions about the use of references, please ask Isabel. You will have to include at least three relevant references related to the topic presented. Please try as much as possible to use sound scientific evidence to support your statements and try to use scientific articles as references.

Please organise the text in a clear, structured way, starting with a short introduction that presents the topic, then going more into details and finally coming up with some conclusions. Keep in mind that this is an academic text, so you should avoid subjective references or personal opinions (as opposed to the reflection exercise, in which we are specifically asking you for your opinion). The idea with this exercise is that you explore a bit more one of the topics presented in the sessions. You will have to read at least three other papers related to the topic (those you will cite in your reference list). This exercise will provide good practice for your reading and writing skills.

## Appendix B

#### Interviews ad-hoc at the event (conducted by three different interviewers)

# Warm-up

- What have you been doing, what are you planning to do/participate in?
- Did you find this event to be interesting? What in it? What is most interesting?

# Knowledge & Expertise

- The aim is to find out 1) what the student's perception of their expertise is and how it is related to the assembly, 2) if they potentially feel connected to the assembly that way and 3) if they are motivated to be part of the groups of experts that are present at the event, and 4) how do they personally relate to the Arctic area.
  - What are you studying (in EnCHiL or in landscape restoration)?
  - Do you have any plans after graduating?
  - How familiar are you with the themes of this event? Are you studying these themes?
  - What is your expertise? Is it represented in this event?
  - Are the themes of the event part of your career plans?
  - Are you from the Nordics/Arctic? What's your relation to the area?

## Social interaction

- Aim is to find out 1) who the student has interacted with 2) who they would like to interact with and 3) how "socially orientated" they are towards the assembly.
  - Have you talked with people here?
  - Are you going around alone or in groups? Are you looking to network with people?
  - Who are you planning to talk to for your exercise? Why?

# 730 Learning

735

740

720

- Do you feel like you are learning here? What?

# **Belonging**

Aim is to find out 1) do the students feel welcome and comfortable in the course or at the Assembly, 2) what are the groups that they associate themselves with (with using the information gained in the previous parts of the interview, 3) where the students potentially feel like belonging to.

- Did you feel welcome to the course? What created it?
- Did you feel welcome at the Assembly? What created it?
- What group do you belong to at this event?
- Do you feel like a part of the expert community here? Do you see yourself part of that in the future?
- Do you feel like a part of the student group here?
  - Do you feel like a part of the.... whichever group has come up in the discussion.
  - Do you feel like you belong here? What is 'here' and how do you feel to be a part of it?
  - Do you feel like an outsider here? Why?

## 745 Post-interviews (conducted by one interviewer)

#### What did you learn?

- Change in existing knowledge
- Various things such as skills, knowledge, about the world, about arctic...

# Did something transform?

- 750 Knowing differently now
  - o How did it change how you understand the themes or how you feel, think or behave?
  - Relating to themes/concepts/people differently
    - o Perceiving your own role or responsibility differently?
    - o Sense of own expertise—Feeling more like an expert now?
  - Intention to seek for transformation?

# How do you see your sense of belonging to connect to the transformative learning/experience during the course?

- How did your sense of belonging change during the experience?
- A connection between your (transformative) learning and sense of belonging
- Did it enable or reversely-enable?

755