# Peer review of "Seeds of transformative learning and its pedagogical implications on a conference-based university course for environmental and geosciences"

_EGUsphere, 2024_

## Author Response (AR1)

We thank both the referees for their insightful comments and suggestions! We have taken them into account and edited the manuscript accordingly. Specific changes are collected in the table below.

Best regards, On behalf of all the co-authors Joula Siponen

| REFEREE | # | Section/location
(lines in the old
version) | Referee comment                                                                                                                                                                                                                                                                                                                                                                                                                                                              | Authors' response                                                                                                                                                                                                                                                                                                    | Change in the manuscript (lines in the revised, clean version)                                                                                                                                                                                                                        |
|---------|---|---------------------------------------------------|------------------------------------------------------------------------------------------------------------------------------------------------------------------------------------------------------------------------------------------------------------------------------------------------------------------------------------------------------------------------------------------------------------------------------------------------------------------------------|----------------------------------------------------------------------------------------------------------------------------------------------------------------------------------------------------------------------------------------------------------------------------------------------------------------------|---------------------------------------------------------------------------------------------------------------------------------------------------------------------------------------------------------------------------------------------------------------------------------------|
| #1      | 1 | Introduction, lines 64
to 67                   | I think the fact that scientists working on climate change have no training in communication their complex results to a wide audience is a very important topic here. The role of scientific organisation, such as the IPCC, as a bridge between scientific community and policy makers and stake holders could be mentioned as a potential bridge. For popular science, scientists are also not alone and many organisations or associations help bridge the knowledge gap. | Thank you for the comment and highlighting the existing bridges for communication of geoscience to the public. You are right that there are channels already available and people dedicated to science communication within various organisations. It is important to acknowledge those structures.                  | Added sentence, line 68-70: 'Structures, such as the Intergovernmental Panel on Climate Change (IPCC), hold an important position in supporting communication of geosciences to the public and policymaking, and working as a bridge between the science community and stakeholders.' |
|         | 2 | Introduction, lines 81
and forward             | There is a 1.1 but no 1.2. This section looks a little bit like a "Methods" section, in the sense that it explains how the study was built.                                                                                                                                                                                                                                                                                                                                  | The intention of this section was to describe the course as the setting of our study, and therefore we decided to place it under the introduction. It is true that the graph of the course schedule also includes the steps of the study, referring to methodology. We have now moved the section under methodology. | New name of the section:
3.1 "Arctic Circle" course                                                                                                                                                                                                                                |

|    | 3 | Discussion, lines 455
to 457 | I wonder if any of the interview targets why they think so few climate change expert actually take these steps currently? Or society in general.                                                                                                                                                                                                                                                                                                                                                                                           | This is an interesting question! The views of the students on why these steps are rarely taken was not directly discussed in our interviews. However, it would be interesting to further investigate the views and attitudes of geoscience researchers and students towards responsible scientific practice, climate justice and social sustainability.     |                                                                                                                                                                                                                     |
|----|---|---------------------------------|--------------------------------------------------------------------------------------------------------------------------------------------------------------------------------------------------------------------------------------------------------------------------------------------------------------------------------------------------------------------------------------------------------------------------------------------------------------------------------------------------------------------------------------------|-------------------------------------------------------------------------------------------------------------------------------------------------------------------------------------------------------------------------------------------------------------------------------------------------------------------------------------------------------------|---------------------------------------------------------------------------------------------------------------------------------------------------------------------------------------------------------------------|
| #2 | 4 | Introduction, lines 52-53       | The Introduction is well-written and provides the necessary information about the topic. In lines 52-53, the emphasis on a transdisciplinary approach is well-founded, however, the manuscript could benefit from clarifying how transdisciplinarity is operationalized in practice.                                                                                                                                                                                                                                                       | We added a sentence in the manuscript.                                                                                                                                                                                                                                                                                                                      | Added sentence, line 53-55: 'Being transdisciplinary in climate education is often operationalised in problemand/or phenomena-based learning, when students are exposed to real-world problem contexts and events.' |
|    | 5 | Introduction, lines 68-
74   | Although the authors state what they do in their study, this does not in itself suffice as a rationale for the importance of the study; it is merely procedural. That is, it tells what the authors are doing, but not why it matters. To be a solid rationale, the authors should move beyond description and state the importance of the problem, explain how the study may affect the current situation and the relevant research approaches that are being used at the moment and state the originality and contribution of the study. | Our work aims to open a new strand of discussion on sense of belonging and transformative learning specifically in the context of environmental and geoscience higher education. That is highly needed as future experts and professionals in climate and environmental change have been shown to require reflexive capacity and transformative competence. | Added a sentence to the beginning (line 71) and to the end (line 79) of the paragraph to better justify the importance.                                                                                             |

| 6 | Introduction, RQs                                     | I would strongly advise the authors to reword the first research question with greater clarity and to break the second question into two questions as the second research question should not be that broad.                                                                                               |                                                                                                                                                                                                                                                                                                                                                                                                                                                                                                                                                                                                      | Lines 82-85: The first question is clarified and the second question is divided into two parts.                      |
|---|-------------------------------------------------------|------------------------------------------------------------------------------------------------------------------------------------------------------------------------------------------------------------------------------------------------------------------------------------------------------------|------------------------------------------------------------------------------------------------------------------------------------------------------------------------------------------------------------------------------------------------------------------------------------------------------------------------------------------------------------------------------------------------------------------------------------------------------------------------------------------------------------------------------------------------------------------------------------------------------|----------------------------------------------------------------------------------------------------------------------|
| 7 | Introduction, Lines 78-
80                         | The information should be more analytical                                                                                                                                                                                                                                                                  | We formulated the paragraph to have more of an analytical tone to better reflect the research process.                                                                                                                                                                                                                                                                                                                                                                                                                                                                                               | Lines 86-89.                                                                                                         |
| 8 | Course description
(Methodology), lines
109-114 | It provides key information about the transformative component of the course and as such it should be much more analytical; it should account for how exactly the course was transformative. Therefore, very small sentences should be avoided and a more explanatory way of expression should be adopted. | The course was not designed specifically with transformative learning in mind and our main interest was to study whether such learning can be spotted. In this section we wanted to compare the pedagogical setting to what has been recently suggested as pedagogical elements that can support transformative learning. Therefore, we hesitate to say here explicitly what makes the course transformative but instead speculate the suitability of the course design for such learning to emerge. We will clarify this in the text and organise the elements as a list with widened explanations. | Lines 190-200: clarification and a list added                                                                        |
| 9 | Course description
(Methodology), section
1.1   | There are some concerns about the structure of the paper; it would be more suitable to place Section 1.1 (with the right adjustments) within the Methodology section and not before the literature section. I would also advise omitting "Case" from the heading of the section.                           | We made changes accordingly as a similar comment was received from the first referee.                                                                                                                                                                                                                                                                                                                                                                                                                                                                                                                | Section 3 heading now Methodology, and under it 3.1 "Arctic Circle" course, 3.2 Materials, and 3.3 Analysis process. |

| 10 | Theoretical framing, section 2        | In addition, a more suitable heading for section "2
Theoretical framing" would be "Theoretical
background".                                                                                                                                                                                                                                                                                                                                                             |                                                                                                                                                                                                                                                                                                                                                                                                                      | Section 2 heading now
Theoretical background                                                                                                                                                                                   |
|----|---------------------------------------|-------------------------------------------------------------------------------------------------------------------------------------------------------------------------------------------------------------------------------------------------------------------------------------------------------------------------------------------------------------------------------------------------------------------------------------------------------------------------------|----------------------------------------------------------------------------------------------------------------------------------------------------------------------------------------------------------------------------------------------------------------------------------------------------------------------------------------------------------------------------------------------------------------------|-----------------------------------------------------------------------------------------------------------------------------------------------------------------------------------------------------------------------------------|
| 11 | Theoretical framing,
lines 127-129 | The learning outcomes ought to be explained more and worded with greater clarity.                                                                                                                                                                                                                                                                                                                                                                                             | We further clarified what is meant with each transformative learning outcome and listed them more clearly as themes.                                                                                                                                                                                                                                                                                                 | Lines 99-105: clarified list of learning outcomes                                                                                                                                                                                 |
| 12 | Theoretical framing, section 2        | it is necessary to improve the coherence and flow between paragraphs. At times, transitions are abrupt, and the logical progression of ideas is unclear. Strengthening the internal linking between paragraphs will help guide the reader through the argument more effectively. Additionally, some key claims would be more convincing if supported by further elaboration or evidence. I also recommend conducting a more extensive literature review and add more sources. | We utilised a suitable theoretical frame for our analysis and study and aimed to introduce the topic through that framing, rather than conducted a literature review on transformative learning extensively.                                                                                                                                                                                                         | Edits to the wording to highlight the linkages between the all paragraphs in section 2 (e.g. lines 108, 121, 131). Claridfied that the primary source of framing was a review by Rodríguez Aboytes and Barth (2020) (line 97-99). |
| 13 | Methodology, section 3                | Given the very low number of participants, it is necessary to explain how thematic saturation was achieved and ensured in the semi-structured interviews before and after the course.                                                                                                                                                                                                                                                                                         | Our methodological emphasis was on representing the perspectives of the study cohort (course participants) rather than seeking a thematic saturation per se – thus aiming at depth rather than width in the material. We addressed the potential limitations of the sample (and of course the study in general) in the methodological chapter and in a dedicated paragraph of limitations in the Discussion chapter. | Small additions to methods (e.g. lines 205-208) and discussion (lines 502-505).                                                                                                                                                   |

| 14 | Methodology, section 3   | Regarding reflections, although they should be included, it is necessary to explain how they were systematically collected and analyzed (as for a recognized qualitative method).                                                                                                                                                                                                                                                                                                                                                                                          | The written reflections were part of the mandatory course assignments, and the teacher of the course shared them with the other authors after they were submitted by the students (as declared to the students when introducing the study in the online session prior to the event). Reflections were then analysed together with the interview transcripts following qualitative content analysis process assisted by Atlas.ti program. | Added clarifications to data collection (e.g. line 206->) and analysis (234-235).      |
|----|--------------------------|----------------------------------------------------------------------------------------------------------------------------------------------------------------------------------------------------------------------------------------------------------------------------------------------------------------------------------------------------------------------------------------------------------------------------------------------------------------------------------------------------------------------------------------------------------------------------|------------------------------------------------------------------------------------------------------------------------------------------------------------------------------------------------------------------------------------------------------------------------------------------------------------------------------------------------------------------------------------------------------------------------------------------|----------------------------------------------------------------------------------------|
| 15 | Methodology, section 3.2 | It is concerning that although the first interviews were conducted in person, the in-depth post-course interviews were conducted in Zoom. As in-person interviews allow for richer non-verbal communication and a more natural conversational flow ensuring the understanding of the topic, Zoom interviews may limit observation or introduce technical distractions, possibly influencing the depth of reflection or emotional expression. This is an important flaw and authors should explain how the mode of communication might have shaped participants' responses. | As the students and the researchers were located in different countries, online interviews were the most sensible option post-course. The distance can also be beneficial to the level of reflection as the students may feel more comfortable to speak candidly online. This technical detail was not seen to influence the communication and thus the results of the study significantly.                                              | Added short reflection (lines 219-223).                                                |
| 16 | Methodology, section 3.2 | In the beginning of the section, it is necessary to state explicitly the type of interviews that were conducted before and after the course and to state the reasons for choosing these methods and most importantly explain how these methods served better the research aim.                                                                                                                                                                                                                                                                                             | Qualitative and explorative methodology was seen relevant for gaining new insight on the students' personal learning experiences. Semi-structured interviews in-situ and post-fact, supported by written reflections provided rich and detailed data.                                                                                                                                                                                    | A more clear and explicit statement is added to the beginning of section (line 205->). |

| 1  | 7 Methodology, section 3.3 | There should also be a methodological framing of the approach used to analyze interview data. In addition, it is necessary to explain how coding was done and if a framework was used and how themes were exactly developed.                                                            |                                                                                                                                                                                                                                                                                                                       | Section 3.3: Explained the process more clearly and referenced qualitative content analysis in general by Bryman (2016) and for the reappearances in grouping the findings as suggested by                                                   |
|----|----------------------------|-----------------------------------------------------------------------------------------------------------------------------------------------------------------------------------------------------------------------------------------------------------------------------------------|-----------------------------------------------------------------------------------------------------------------------------------------------------------------------------------------------------------------------------------------------------------------------------------------------------------------------|----------------------------------------------------------------------------------------------------------------------------------------------------------------------------------------------------------------------------------------------|
| 1  | B Discussion, section 5    | it is necessary to state concisely whether and how the study managed to answer the research questions. The Discussion could also provide some critique on previous relevant studies and point at how the study challenges or advances the approaches that have been followed until now. |                                                                                                                                                                                                                                                                                                                       | Krippendorff (2019).  A clearer statement of our reflections on the success of the research design, highlighted key findings, and how the research questions were answered, in added now in the beginning of the Discussion (lines 445-451). |
| 19 | 9 Discussion/Conclusion    | The paper would be further improved if the authors provided a separate Conclusions section where the authors will summarize key conclusions, reiterate the significance of the study, state study limitations and suggest directions for future studies in this research field.         | We consider the last paragraph of the Results and Discussion chapter to be the conclusive chapter of our submission—and to stylistically function as is, without a dedicated subheading. We hope this is a satisfactory way of structuring the manuscript and to be in accordance with the guidelines of the journal. | The beginning of the paragraph is changed to 'In conclusion' for clarity.                                                                                                                                                                    |
| 20 | 0 General                  | I would also advise the authors to perform language editing to ensure accuracy and clarity and to avoid any typos both in the paper and the Appendices.                                                                                                                                 |                                                                                                                                                                                                                                                                                                                       | Language check conducted with small corrections throughout the manuscript.                                                                                                                                                                   |
| 2  | 1 Abstract                 | After revising the paper, it would be beneficial to revise also the Abstract so that it becomes more engaging and accurate.                                                                                                                                                             |                                                                                                                                                                                                                                                                                                                       | Edited the order of sentences, starting more clearly with what has been done in this study.                                                                                                                                                  |

|               | 22 | Technical: Abstract, lines 16-17    | it would read better to replace "students' learning experience on a university course" with "students'                                                                                          | Changed as suggested.                                                                                                                                                                                                |
|---------------|----|-------------------------------------|-------------------------------------------------------------------------------------------------------------------------------------------------------------------------------------------------|----------------------------------------------------------------------------------------------------------------------------------------------------------------------------------------------------------------------|
|               |    | unes 10-17                          | learning experience during a university course".                                                                                                                                                |                                                                                                                                                                                                                      |
|               | 23 | Technical: Abstract,
lines 16-17 | it would also be better to replace "where students of environmental and geosciences attend" with something a bit clearer such as "where students studying environmental and geosciences attend" | Changed as suggested with language edits.                                                                                                                                                                            |
|               | 24 | Technical: Abstract                 | instead of "With a qualitative inquiry", it would be perhaps more accurate to stat "Using qualitative methods".                                                                                 | Changed as suggested.                                                                                                                                                                                                |
|               | 25 | Technical: Line 54                  | To improve clarity, I would advise to remove sentence "This poses a challenge for geoscience education".                                                                                        | Line 56: Removed the sentence                                                                                                                                                                                        |
|               | 26 | Technical: Line 58                  | Remove "s" from grasps in "The experts in Arctic climate change must grasps".                                                                                                                   | Changed as suggested.                                                                                                                                                                                                |
|               | 27 | Technical: Line 89                  | Instead of providing the link as citation, perhaps it would be better to provide the name of the institution and a date - if available).                                                        | Changed as suggested, added to the reference list.                                                                                                                                                                   |
|               | 28 | Technical: Lines 100-
102        | Sentence ought to be rephrased as it is somewhat confusing.                                                                                                                                     | Changed the sentence to 'Before the conference, each student presented their participation plans in an online session, allowing their peers to be motivated by the topics and activities chosen by fellow students.' |
|               | 29 | Technical: Table 1                  | The caption for Table 1 should appear above and not below it.                                                                                                                                   | Changed as suggested.                                                                                                                                                                                                |
| Other changes | 30 | Acknowledgements                    |                                                                                                                                                                                                 | Updated.                                                                                                                                                                                                             |